# Extraction of Bioactive Compounds and Influence of Storage Conditions of Raw Material *Chamaenerion angustifolium* (L.) Holub Using Different Strategies

**DOI:** 10.3390/molecules29235530

**Published:** 2024-11-22

**Authors:** Domantas Armonavičius, Mantas Stankevičius, Audrius Maruška

**Affiliations:** Instrumental Analysis Open Access Centre, Institute of Research of Natural and Technological Sciences, Faculty of Natural Sciences, Vytautas Magnus University, Vileikos St. 8, LT-44404 Kaunas, Lithuania; domantas.armonavicius@vdu.lt (D.A.); mantas.stankevicius@vdu.lt (M.S.)

**Keywords:** biologically active compounds, extraction, phenolic compounds, ellagitannins, oenothein B, antioxidants, HPLC, electrochemical detection

## Abstract

The study evaluates different preparation methods for identifying the best strategy for extracting biologically active compounds from raw *Chamaenerion angustifolium* (L.) Holub plant material. The methodologies include direct aqueous methanol extraction with a combination of natural aerobic and anaerobic fermentation for 24–72 h, followed by 35 °C and 60 °C drying. Furthermore, the study also focuses on determining the different temperature storage conditions on the stability of biologically active compounds. UV-VIS spectroscopy was used to quantitatively evaluate the total content of phenolic compounds, flavonoids, and radical scavenging activity. For qualitative analysis, chromatographic separation with electrochemical detection (ED) of extracted compounds, a gradient high-performance liquid chromatography (HPLC) system was used. Study results indicate that 48 h natural aerobic fermentation followed by 35 °C drying and 75% (*v*/*v*) aqueous methanol extraction yielded the maximum amount of biologically active compounds in *Chamaenerion angustifolium* (L.) Holub leaves, blossom, and stem samples. Freezing samples in liquid nitrogen had the lowest impact on the total content of phenolic compounds, flavonoids, and radical scavenging activity. HPLC-ED system results identified chlorogenic acid, oenothein B, trans-p-Coumaric acid, ellagic acid, and rutin in *Chamaenerion angustifolium* (L.) Holub leave samples.

## 1. Introduction

The search for medicinal plants as a source of biologically active compounds has been a considerable part of traditional medicine across different cultures. The diverse chemical composition of these plants yields a significant amount of biologically active compounds with potential health benefits. Due to their safety, effectiveness, therapeutic potential, and minimal side effects, medicinal plants have been of interest in many scientific, pharmaceutical fields, and food industries for many decades [1,2,3]. Although the traditional use of medicinal plants has been well researched, the interest in enhancing different extraction techniques and storing conditions impacts the quantity of biologically active compounds still present to this day.

Scientific literature highlights the diversity of used methods in extracting biologically active compounds from raw medicinal plants [4]. Yet, a comprehensive understanding of the comparative effectiveness of different extraction strategies, including direct extraction of biologically active compounds using solvents of different polarity, natural aerobic and anaerobic fermentation, and the impact of storage conditions on the quantity of these compounds, remains challenging.

Fireweed, *Chamaenerion angustifolium* (L.) Holub (also called *Chamerion angustifolium* and *Epilobium angustifolium*), is a well-known medicinal plant found in many countries of the Northern Hemisphere, including Lithuania. Like any other medicinal plant, *C. angustifolium* has an abundance of polyphenols, especially ellagitannins, flavonoids, and phenolic acids [5]. Research on *C. angustifolium* raw materials has received considerable attention in the last twenty years. At the beginning of the century, *C. angustifolium’s* antiproliferative properties against human prostate cancer were discovered [6,7]. The primary effect is attributed to oenothein B, a compound belonging to the group of oligomeric ellagitannins [8]. Our previous research has shown that *C. angustifolium* extracts inhibit MCF7 and MDA-MB-468 cell line growth [9]. Some studies demonstrate that *C. angustifolium* has a positive effect on benign prostatic hyperplasia (BPH) and potentially on prostate cancer chemoprevention [10]. One study demonstrated that *C. angustifolium* extracts have valuable ingredients to be used in cosmetic and dermatological products [11].

This study aims to systematically explore and analyse the impact of freezing, drying, direct extraction, and fermentation techniques as preparation methods for raw materials of *C. angustifolium* to achieve maximum extraction of biologically active compounds. Through a large number of samples and combinations, we seek to contribute insights that can be used to develop standardized and efficient practices in different industries and scientific fields.

## 2. Results

### 2.1. Fresh and Dried Plant Samples

After separating all raw *C. angustifolium* material (Group 1 samples), plant extraction was started within 2–4 h. After 24 h extraction, the total content of phenolic compounds, flavonoids, and radical scavenging activity was determined. Spectrometric analysis results are presented in Table 1.

Statistical analysis indicated that an increasing MeOH concentration, while extracting biologically active compounds, significantly (*p* < 0.001) affects the total content of phenolic compounds, flavonoids, and radical scavenging activity within all plant samples. There are significant differences in biologically active compound levels across different plant parts (leaves, blossoms, and stems).

The biggest concentration of phenolic compounds, measured at 396.82 ± 15.41 RE mg/g, was determined in the fresh leaves sample through 75% (*v*/*v*) aqueous methanol extraction. In contrast, the lowest concentration of phenolic compounds was determined in the aqueous extraction of fresh stems sample, measuring only 22.52 ± 1.01 RE mg/g. The biggest concentration of flavonoids was determined in the fresh blossoms sample using 75% (*v*/*v*) aqueous methanol extraction (57.19 ± 2.44 RE mg/g). In contrast, the lowest concentration was determined to be in the aqueous extraction of fresh stems sample (3.22 ± 0.09 RE mg/g). The biggest antioxidant activity was measured in the fresh leaves sample using 75% (*v*/*v*) aqueous methanol extraction (359.56 ± 12.99 RE mg/g). In contrast, the lowest activity was determined in the aqueous extraction of fresh stems sample (15.28 ± 0.52 RE mg/g).

The total content of phenolic compounds, flavonoids, and radical scavenging activity of Group 2 samples (plant parts that were air-dried at room temperature for 7 days) is presented in Table 2.

It can be observed that 7-day air-drying at room temperature has a negative effect on the total content of biologically active compounds found in different plant parts. The data correspond with the findings of other studies, which compared the total phenolic compounds content, total flavonoid content, and radical scavenging activity of fresh and dried medicinal herbs [12,13]. When comparing Group 1 and Group 2 aqueous and 75% (*v*/*v*) aqueous methanol extraction results, drying plant parts drastically decreases the total content of phenolic compounds by 121.22–291.60 ± 4.02 RE mg/g in leaf samples. A similar decrease was determined in blossom samples, where polyphenols decreased by 92.25–253.46 ± 3.11 RE mg/g. The lowest decrease of phenolic compounds was determined in stem samples, where they decreased by 11.34–19.49 ± 0.44 RE mg/g. Similar extraction results can be observed in the total content of flavonoids, while in leaf samples, they decreased by 7.55–29.74 ± 0.59 RE mg/g, in blossom samples by 8.81–34.55 ± 0.69 RE mg/g, and in stem samples by 0.62–1.28 ± 0.18 RE mg/g. The most substantial effect of 7-day room drying was seen on radical scavenging activity, whereas in leaf samples, it decreased by 100.30–272.71 ± 2.86 RE mg/g, in blossom samples by 91.90–233.65 ± 2.79 RE mg/g, and in stem samples by 7.69–16.33 ± 0.49 RE mg/g. The total content of phenolic compounds, flavonoids, and radical scavenging activity in air-dried leaf samples was reduced by 67.9–73.5%, 41.3–62.9%, and 66.9–75.5%, respectively. Similar biologically active compound decreasing results were observed in blossom samples, whereas levels reduced by 72.0–78.7%, 54.4–61.7%, and 79.2–82.5%, respectively. The lowest decrease was observed in stem samples, where total phenolic compounds, flavonoids, and radical scavenging activity decreased by 50.4–57.5%, 15.2–34.7%, and 50.4–61.8%, respectively. 75% (*v*/*v*) aqueous methanol extraction yielded the biggest amount of biologically active compounds from air-dried *C. angustifolium* material. After running statistical analysis, Group 1 and Group 2 biologically active compound levels’ results were statistically significant (*p* < 0.001).

### 2.2. Fermentation Impact on Biologically Active Compounds

#### 2.2.1. Aerobic Fermentation

The effects of 24, 48, and 72 h of natural aerobic fermentation following the drying process of the plant parts at 35 °C and 60 °C temperatures on the content of bioactive compounds can be seen in Table 3.

The results of natural aerobic fermentation impact on total biologically active compound content found in different *C. angustifolium* parts are compared to Group 2 samples (air-dried samples). Statistical analysis showed that there was a significant difference between fermentation time and drying temperature (*p* < 0.004), as well as fermentation time and plant part (*p* < 0.001). Using Tukey’s method, it was found that there was no statistically significant total content of phenolic compounds results between 24 h and 48 h fermentation (*p* > 0.005), although 24–48 h fermentation time compared to 72 h fermentation showed significant differences (*p* < 0.004). When comparing the total content of flavonoids and radical scavenging activity results between 24, 48, and 72 h fermentation, all results were significant (*p* < 0.05).

In comparison with air-dried *C. angustifolium* samples, the results show that the total content of phenolic compounds increased by 64.89 ± 5.94 RE mg/g (reaching 129.99 ± 5.94 RE mg/g) in leaf samples using 48 h natural aerobic fermentation followed by 35 °C drying. In contrast, using 60 °C drying, the total content of phenolic compounds increased only by 28.94 ± 3.41 RE mg/g (reaching 94.04 ± 3.41 RE mg/g). In blossom samples, phenolic compounds increased by 41.82 ± 2.23 RE mg/g (reaching 90.09 ± 2.23 RE mg/g) using 35 °C drying and 15.95 ± 2.69 RE mg/g (reaching 64.22 ± 2.69 RE mg/g) using 60 °C drying. Although the biggest increase in the percentage growth was attained through 25% (*v*/*v*) aqueous methanol extraction, using 75% (*v*/*v*) aqueous methanol extraction in both leaf and blossom samples resulted in the highest quantities of phenolic compounds, reaching 172.95 ± 4.69 RE mg/g and 139.55 ± 5.91 RE mg/g, respectively. In stem samples, the total content of phenolic compounds after 48 h natural aerobic fermentation increased by 18.66 ± 1.12 RE mg/g (reaching 33.07 ± 1.12 RE mg/g) using 35 °C and by 13.38 ± 1.29 RE mg/g (reaching 28.29 ± 1.29 RE mg/g) using 60 °C drying. For both samples, maximum efficiency was achieved using 75% (*v*/*v*) aqueous methanol extraction. However, some study results show that using 72 h natural aerobic fermentation followed by 60 °C temperature drying decreases the total content of phenolic compounds in leaves, blossoms, and stems samples.

In general, natural aerobic fermentation also has a positive impact on the total content of flavonoids found in different *C. angustifolium* parts. The best method to recover the highest amounts was determined using 48 h fermentation followed by 35 °C drying. In such cases, flavonoids in leaf samples increased by 18.05 ± 0.41 RE mg/g (reaching 30.30 ± 0.41 RE mg/g) and by 20.48 ± 1.24 RE mg/g (reaching 31.22 ± 1.24 RE mg/g) in blossom samples. The highest content of flavonoids was observed in 75% (*v*/*v*) aqueous methanol extracts, reaching 36.53 ± 0.83 RE mg/g and 37.98 ± 1.37 RE mg/g, respectively, in leaf and blossom samples. In stem samples, the flavonoid content was low, but 48 h natural aerobic fermentation followed by 35 °C drying and 75% (*v*/*v*) aqueous methanol extraction still had some positive impact and increased the levels by 4.88 ± 0.19 RE mg/g (reaching 9.35 ± 0.19 RE mg/g).

Compared to air-dried samples, natural aerobic fermentation has a positive impact on radical scavenging activity in both leaf and blossom samples. Similarly, 48 h fermentation followed by 35 °C drying and 75% (*v*/*v*) aqueous methanol extraction increased the antioxidant activity by 73.22 ± 4.69 RE mg/g (reaching 160.06 ± 4.69 RE mg/g) in leaf samples. A less positive effect was seen after using 60 °C drying, where activity increased by 44.88 ± 5.91 RE mg/g (reaching 131.72 ± 5.91 RE mg/g). Radical scavenging activity in blossom samples was also seen higher after natural aerobic fermentation, where activity increased by 57.66 ± 4.91 RE mg/g (reaching 121.59 ± 4.91 RE mg/g) after 35 °C drying and by 42.73 ± 3.19 RE mg/g (reaching 106.66 ± 3.19 RE mg/g) after 60 °C drying and 75% (*v*/*v*) aqueous methanol extraction. Interestingly, in some cases, radical scavenging activity in stem samples decreased after 24 h and 72 h natural aerobic fermentation.

#### 2.2.2. Anaerobic Fermentation

The effects of 24, 48, and 72 h of natural anaerobic fermentation following the drying process of the plant parts at 35 °C and 60 °C temperatures on the content of bioactive compounds can be seen in Table 4.

The results of natural anaerobic fermentation impact on total biological active compound content found in different medicinal plant parts are also compared to Group 2 samples. Similar to aerobic fermentation, statistically significant results were observed in anaerobic fermentation (*p* < 0.005).

In comparison with air-dried samples, study results show that after 48 h of natural anaerobic fermentation followed by 35 °C drying, the total content of phenolic compounds increased by 53.95 ± 4.19 RE mg/g (reaching 119.05 ± 4.19 RE mg/g) in leaf samples. The number of phenolic compounds was lower after 60 °C drying and increased only by 12.74 ± 2.48 RE mg/g (reaching 77.84 ± 2.48 RE mg/g). In blossom samples, the extraction of phenolic compounds after the same fermentation and drying conditions was lower. The amount increased by 31.89 ± 3.64 RE mg/g (reaching 80.16 ± 3.64 RE mg/g) after 35 °C and 4.69 ± 1.66 RE mg/g (reaching 40.51 ± 1.66 RE mg/g) after 60 °C drying. Similarly to aerobic fermentation, the biggest percentage increase was observed in the 25% (*v*/*v*) aqueous methanol extraction method. The best method to recover the highest amounts of phenolic compounds from all *C. angustifolium* samples was achieved using 75% (*v*/*v*) aqueous methanol extraction after a 48 h fermentation followed by a 35 °C drying process. After a 48 h fermentation followed by a 35 °C and 60 °C drying process, in stem samples, the total phenolic content reached 30.55 ± 1.22 RE mg/g and 26.99 ± 1.31 RE mg/g, respectively.

Natural anaerobic fermentation had a lower positive impact than aerobic fermentation, but it still acted beneficial by increasing flavonoids in all three different *C. angustifolium* parts. Study results show that the maximum efficiency in extracting these compounds (percentage growth in the number of flavonoids from fresh samples) was achieved after 48 h fermentation followed by 35 °C drying. In leaf samples, the total content of flavonoids increased by 12.95 ± 0.64 RE mg/g (reaching 25.20 ± 0.64 RE mg/g). Slightly lower results were identified after 60 °C drying, where levels increased by 7.33 ± 0.58 RE mg/g (reaching 19.58 ± 0.58 RE mg/g). Similar results of flavonoid increase were also seen in blossom samples, where levels increased by 9.39 ± 0.88 RE mg/g (reaching 25.96 ± 0.88 RE mg/g) and 4.24 ± 0.69 RE mg/g (reaching 20.81 ± 0.69 RE mg/g) after 35 °C and 60 °C drying, respectively. Plant stems do not have many flavonoids in their tissues and cells, so natural anaerobic fermentation has very little effect on increasing the quantities of these compounds. Still, after 48 h of fermentation followed by 35 °C drying, the amount increased by 0.88 ± 0.21 RE mg/g (reaching 4.33 ± 0.21 RE mg/g) and after 60 °C drying by 1.54 ± 0.10 RE mg/g (reaching 3.95 ± 0.10 RE mg/g). Despite these results, while using 25% (*v*/*v*) aqueous methanol extraction, showed the largest percentage growth in all *C. angustifolium* parts, using 75% (*v*/*v*) aqueous methanol extraction was still a better method to achieve flavonoid recovery in leaves, blossom, and stems samples (30.27 ± 1.02 RE mg/g, 32.84 ± 1.29 RE mg/g, and 6.31 ± 0.21 RE mg/g, respectively).

Natural anaerobic fermentation, in most cases, has a positive impact on increasing the radical scavenging activity in *C. angustifolium* samples. Similar to other results, the biggest increase is seen after 48-h natural anaerobic fermentation followed by 35 °C drying and 75% (*v*/*v*) aqueous methanol extraction. In leaf and blossom samples, the antioxidant activity increased by 55.37 ± 6.81 RE mg/g (reaching 142.21 ± 5.81 RE mg/g) and 32.66 ± 4.56 RE mg/g (reaching 96.59 ± 4.56 RE mg/g). Results after 60 °C drying were significantly lower in leaf and blossom samples, resulting in increments of 8.24 ± 3.09 RE mg/g (reaching 95.08 ± 3.09 RE mg/g) and 17.50 ± 3.44 RE mg/g (reaching 81.43 ± 3.44 RE mg/g). Similar to aerobic fermentation impact, from the start of 24 h natural anaerobic fermentation, radical scavenging activity in stem samples decreased and maintained such a trend till the end of a 72 h fermentation.

### 2.3. Freezing Influence on Biologically Active Compounds

#### 2.3.1. Effect of 3 Months of Freezing

Table 5 presents the influence of freezing temperatures of −18 °C, −80 °C, and −196 °C for 3 months on the total content of phenolic compounds, flavonoids, and radical scavenging activity.

The results of the 3-month freezing temperature impact on total biological active compound content found in medicinal plant parts are compared to Group 1 samples. Statistically significant results were found comparing storage temperature conditions (*p* < 0.004) between Groups 3–5.

Samples that were frozen in a common freezer (−18 °C) had the most significant decrease of biologically active compounds in all plant parts compared to deep-freezing (−80 °C) and liquid nitrogen (−196 °C) freezing methods. The total content of phenolic compounds in leaf samples decreased by 22.7–28.1% (50.10–89.89 ± 7.12 RE mg/g), in blossom samples by 26.9–31.2% (36.19–100.15 ± 6.07 RE mg/g), and in stem samples by 22.5–24.1% (5.06–8.17 ± 1.25 RE mg/g). The total content of flavonoids in leaf samples decreased by 22.7–32.8% (5.12–15.52 ± 1.84 RE mg/g), in blossom samples by 22.5–30.1% (4.50–17.23 ± 1.74 RE mg/g), and in stem samples by 28.6–32.7% (1.05–1.61 ± 0.21 RE mg/g). Radical scavenging activity decreased similarly to polyphenolics, where in leaf samples it decreased by 24.5–26.0% (38.95–72.15 ± 5.54 RE mg/g), in blossom samples by 24.4–25.5% (28.33–77.48 ± 6.18 RE mg/g), and in stem samples by 23.8–28.5% (3.96–6.87 ± 1.00 RE mg/g).

The impact of deep-freezing temperatures (−80 °C) on the total content of biologically active compounds’ stability in *C. angustifolium* samples was better when compared to those of the −18 °C freezing method. The total content of phenolic compounds decreased by 10.4–15.7% (20.37–53.70 ± 7.88 RE mg/g) in leaf samples, 11.3–16.3% (19.05–38.65 ± 6.79 RE mg/g) in blossom samples, and 11.4–14.3% (3.04–3.89 ± 1.45 RE mg/g) in stem samples. The total content of flavonoids showed a similar percentage decrease, where in leaf samples, the number of flavonoids decreased by 13.8–17.9% (4.58–8.13 ± 1.89 RE mg/g), in blossom samples by 11.1–13.7% (3.41–8.76 ± 3.05 RE mg/g), and in stem samples by 13.1–17.0% (0.21–1.33 ± 0.22 RE mg/g). Radical scavenging activity decreased the most, where in leaf samples, the activity decreased by 13.7–19.0% (36.37–42.20 ± 7.88 RE mg/g), in blossom samples by 15.6–18.6% (21.11–47.95 ± 6.19 RE mg/g), and in stem samples by 13.8–18.4% (2.81–4.44 ± 1.01 RE mg/g).

The biologically active compound’s stability was highest when frozen in liquid nitrogen (−196 °C). The total content of phenolic compounds in leaf samples decreased only by 1.7–6.6% (4.08–11.87 ± 3.93 RE mg/g), in blossom samples by 2.4–5.8% (7.40–10.02 ± 3.81 RE mg/g), and in stem samples by 1.5–7.0% (0.39–2.36 ± 0.25 RE mg/g). The total content of flavonoids in the *C. angustifolium* sample decreased very similarly to polyphenols. In leaf samples, it decreased by 2.9–8.7% (0.79–4.13 ± 0.68 RE mg/g), in blossom samples by 4.2–7.3% (0.68–3.96 ± 0.44 RE mg/g), and in stem samples by 1.7–7.1% (0.07–0.31 ± 0.06 RE mg/g). The radical scavenging activity showed similar decreasing results. In leaf samples, it dropped by 2.9–4.8% (6.38–8.03 ± 4.82 RE mg/g), in blossom samples by 1.7–6.5% (1.91–17.85 ± 1.49 RE mg/g), and in stem samples by 3.4–6.7% (0.81–1.62 ± 0.77 RE mg/g).

#### 2.3.2. Effect of 6 Months of Freezing

Table 6 presents the influence of freezing temperatures of −18 °C, −80 °C, and −196 °C for 6 months on the total content of phenolic compounds, flavonoids, and radical scavenging activity.

The results of the 6-month freezing temperature impact on total biologically active compound content found in medicinal plant parts are compared to Group 1 samples. It can be observed that storing medicinal plants at −18 °C, −80 °C, and −196 °C temperatures for 6 months showed a more severe decrease in biological active compound content found in *C. angustifolium* when compared to storing for 3 months. Statistically significant results were found comparing storage temperature conditions (*p* < 0.004) between Groups 3–5 and Groups 6–8.

When comparing the stability of polyphenolic compounds, the most significant negative impact was observed at a −18 °C freezing temperature. The total content of phenolic compounds decreased by 29.4–41.2% (73.58–121.48 ± 5.19 RE mg/g) in leaf samples, 34.7–47.2% (57.68–151.34 ± 4.58 RE mg/g) in blossom samples, and 27.8–35.4% (6.80–11.26 ± 1.01 RE mg/g) in stem samples. In leaf samples, flavonoids decreased by 39.6–46.0% (8.40–19.08 ± 0.99 RE mg/g), in blossom samples by 29.9–40.1% (5.18–20.58 ± 1.45 RE mg/g), and in stem samples by 28.5–27.7% (5.07–14.08 ± 0.51 RE mg/g). Over time, radical scavenging activity decreased slightly less compared to the polyphenolic amount, where in leaf samples, it decreased by 27.6–41.8% (62.70–94.48 ± 4.64 RE mg/g), in blossom samples by 30.2–36.1% (34.27–103.67 ± 4.63 RE mg/g), and in stem samples by 27.4–38.6% (5.45–9.08 ± 0.59 RE mg/g).

Compared to 3-month −80 °C temperature storage results, the stability of biologically active compounds after 6 months was slightly lower. The total content of phenolic compounds decreased by 19.4–26.2% (43.81–82.05 ± 5.22 RE mg/g) in leaf samples, 18.7–29.4% (29.51–94.17 ± 4.96 RE mg/g) in blossom samples, and 18.8–25.9% (5.31–6.88 ± 0.99 RE mg/g) in stem samples. The total content of flavonoids decreased even more after 6-month freezing; in leaf samples, the number of flavonoids decreased by 21.7–29.8% (5.07–14.08 ± 1.45 RE mg/g), in blossom samples by 20.9–24.9% (3.38–14.24 ± 1.59 RE mg/g), and in stem samples by 22.6–26.4% (0.80–1.29 ± 0.19 RE mg/g). Radical scavenging activity had a slightly less negative effect when compared to the total content of phenolic compounds; in leaf samples, levels decreased by 22.4–27.5% (36.54–69.28 ± 5.09 RE mg/g), in blossom samples by 22.9–29.4% (25.96–84.40 ± 5.66 RE mg/g), and in stem samples by 20.8–23.7% (3.49–6.02 ± 0.83 RE mg/g).

Storing *C. angustifolium* in liquid nitrogen had the lowest negative impact on biological active compound stability when compared to −18 °C and −80 °C temperature storage. The total content of phenolic compounds in leaf samples after 6 months decreased only by 10.4–11.3% (19.42–41.18 ± 5.49 RE mg/g), in blossom samples by 10.6–11.5% (17.11–36.83 ± 6.79 RE mg/g) and stem samples by 8.5–9.9% (2.07–3.37 ± 1.02 RE mg/g). The total content of flavonoids in leaf samples decreased by 10.2–11.9% (1.86–5.36 ± 1.68 RE mg/g), in blossom samples by 9.3–14.1% (2.28–7.26 ± 2.12 RE mg/g) and in stem samples by 8.7–12.4% (0.32–0.60 ± 0.22 RE mg/g). The radical scavenging activity also decreased a bit more when compared to the 3-month freezing effect, but the impact was still very low. In plant samples, the antioxidant activity decreased by 10.1–12.6% (17.22–35.11 ± 5.73 RE mg/g), in blossom samples by 10.4–12.6% (14.29–35.28 ± 4.79 RE mg/g) and in stem samples by 8.6–11.6% (1.31–3.18 ± 1.08 RE mg/g).

### 2.4. Quantitative HPLC-ED Analysis

The chromatography system with an electrochemical detector (HPLC-ED system) was used for a reversed-phase liquid chromatography analysis in *C. angustifolium* L. plant extracts. The results of the quantitative analysis are shown in Table 7, Table 8 and Table 9.

It can be observed that out of twelve external standards, only three compounds are identified in dried *C. angustifolium* leaf samples. Additionally, previous studies indicated that using reversed-phase HPLC systems with UV detection identified oenothein B as being eluted the first from the column in high quantities. The results of this study showed the first compound being eluted from the column in high quantities and that was allocated to oenothein B. In total, four compounds were identified using the HPLC-ED system.

HPLC-ED analysis identified that the highest concentration of oenothein B in leaf samples was obtained after air-drying and using 75% (*v*/*v*) aqueous MeOH extraction. The concentration reached 64.36 ± 0.10 RE mg/g. The second highest concentration was identified as chlorogenic acid, being 51.65 ± 0.09 RE mg/g after air-drying samples and using 75% (*v*/*v*) aqueous MeOH extraction. A significantly lower concentration was identified as rutin, being 15.81 ± 0.02 RE mg/g after the same extraction conditions. Only traces of ellagic acid are identified in the leaf samples. Similar to spectrometric analysis, the HPLC-ED system shows significant (*p* < 0.005) results that increasing the concentration of methanol, higher amounts of phenolic compounds are extracted from *C. angustifolium* leaf samples.

After 24–72 h of natural aerobic leaf sample fermentation, followed by a 35 °C drying process, only trans-p-coumaric acid is identified using the HPLC-ED system. Similarly, with spectrometric analysis, HPLC-ED results show that with increasing fermentation time up to 48 h, trans-p-coumaric acid quantity increases to 32.38 ± 0.09 RE mg/g, but after 72 h, it drastically decreases to only traces. A similar trend, but a lower amount of trans-p-coumaric acid, is detected after anaerobic leaf sample fermentation. The quantity of the compound was lower by approximately 20.31 ± 0.06 RE mg/g after 48 h of natural anaerobic fermentation followed by a 35 °C drying process. None of the identified phenolic compounds (oenothein B, chlorogenic acid, rutin, and ellagic acid) was detected after natural aerobic and anaerobic fermentation, which indicates that during fermentation, their electrochemical properties may be affected by various chemical changes, resulting in the HPLC-ED system not detecting these compounds.

After collecting different fresh *C. angustifolium* parts, all samples were kept at −18 °C, −80 °C and −196 °C temperatures for 3 and 6 months. After preservation time, oenothein B, chlorogenic acid, and rutin were identified using the HPLC-ED system. Similar to spectrometric analysis, the same trend can be identified: increasing negative temperatures, phenolic compounds are more stable, and their electrochemical properties in leaf samples are maintained.

## 3. Discussion

The study aimed to determine the efficiency of various extraction methods and the influence of storage conditions on the quantity of biologically active compounds derived from *C. angustifolium*. Comparing extraction methods highlights distinct advantages in each approach, providing valuable insights into optimizing the extraction process.

Our research results reveal that in all cases, the 75% (*v*/*v*) aqueous methanol extraction method yields the highest amount of bioactive compounds. In fresh plant samples, the leaves contained the highest levels of phenolic compounds compared to blossoms, which contained the highest levels of flavonoids. Since radical scavenging activity is mainly determined by the amount of polyphenols present in the plants, the highest antioxidant activity was indeed found in leaf samples. Study results show that air-drying samples for 7 days at room temperature have a drastically negative effect on *C. angustifolium* samples. The total content of phenolic compounds in dried leaf samples decreased up to 73%, while radical scavenging activity decreased up to 68%. The total content of flavonoids in dried blossom samples also decreased significantly—up to 60%. In general, study results showed that stem samples do not have large quantities of biologically active compounds.

The use of both natural aerobic and anaerobic fermentation methods showed a positive impact on increasing the biologically active compounds in *C. angustifolium* samples. Spectrometric analysis results indicate that compound levels in leaves, blossoms, and stems increased from the beginning up to the 48 h fermentation period. However, once the fermentation duration reached 72 h, a decrease in phenolic compounds, flavonoids, and radical scavenging activity was noticeable. While spectrometric analysis results demonstrated a quantitative increase in biologically active compounds, the HPLC-ED system results indicated qualitative changes in extracted compounds. Chromatographic separation with electrochemical detection showed only a few compounds identical to those extracted from unfermented raw material, and their peak area was significantly lower. These results are aligned with some other studies that demonstrate the positive fermentation effect of *C. angustifolium* raw material [13,14,15]. When comparing both natural fermentation methods, it is seen that the use of natural aerobic fermentation resulted in larger quantities of biologically active compounds across all samples. These results indicate that the presence of oxygen provides a more suitable environment for metabolic processes. Furthermore, the usage of either 35 °C or 60 °C temperatures for drying the samples after the fermentation showed significant differences. Results indicate that drying all samples at 35 °C yielded better results in preserving biologically active compounds. This finding indicates that lower temperatures minimise the risk of thermal degradation.

The influence of freezing *C. angustifolium* samples in the common freezer, deep-freezing temperatures, and liquid nitrogen for 3 and 6 months had different noticeable impacts on the stability of biologically active compounds. While all temperatures showed a negative effect, lower temperatures resulted in better stability over time. Such results indicate that freezing samples in liquid nitrogen correlates to the physical and biochemical properties of water, enzymatic activity, and microbiological growth. Once the water in plant cells is rapidly frozen, the hydration state of the sample is prevented, which helps maintain the biochemical profile of the herb. Liquid nitrogen also almost instantly stops the enzymatic activity, preventing the degradation of the biologically active compounds. Low temperatures inhibit microbial growth, preventing fermentation and other microbial processes that might degrade other biologically active compounds.

Using the HPLC-ED system, only four out of twelve standards were identified: chlorogenic acid, trans-p-coumaric acid, ellagic acid, and rutin. The quantitative HPLC-ED system results correlated with qualitative spectrometric analysis results, showcasing similar behaviour on *C. angustifolium* plant leaf samples. According to the literature search, *C. angustifolium* is high in oenothein B, which is easily soluble in polar solvents and thus eluted from the HPLC column first. After obtaining the chromatogram, the first largest peak is assigned to this compound. Other antioxidant compounds were eluted by analysing air-dried, aerobic/anaerobic fermentation, and frozen leaf extract samples chromatograms, but additional reference standards are needed to identify them in further analysis. Along with *C. angustifolium*, there are numerous medicinal plants that are rich in ellagitannins that demonstrate anticancer and antioxidant properties. Future research will aim to determine how local medicinal plants found in Lithuania can be used in cancer treatment.

## 4. Materials and Methods

### 4.1. Plant Material

The raw material of fireweed (*C. angustifolium*) was collected near Kaišiadorys city, Lithuania (54°53′53″ N, 24°29′33″ E) in July 2023. The plant’s raw material was separated into three main parts: leaves, blossoms, and stems. The samples were separated on the same day, a couple of hours after the collection.

### 4.2. Reagents

Methanol (p.a., Chempur, Piekary Śląskie, Poland), sodium carbonate (p.a.; Chempur, Piekary Śląskie, Poland), Folin–Ciocalteu’s phenol reagent (2N; Merck, Darmstadt, Germany), acetic acid (80%; Chempur, Piekary Śląskie, Poland), hexamethylenetetramine (99%; Carl Roth, Karlsruhe, Germany), aluminum chloride (99.0%; Alfa Aesar, Karlsruhe, Germany), sodium acetate (99%; Alfa Aesar, Karlsruhe, Germany), DPPH (2,2-diphenyl-1-picrylhydrazyl) (99%; Sigma Aldrich, St. Louis, MO, USA), acetonitrile (99%; Merck, Darmstadt, Germany), rutin (95%; Merck, Darmstadt, Germany), gallic acid (95%, Thermo Scientific, Waltham, MA, USA), vanillic acid (97%, Sigma Aldrich, St. Louis, MO, USA), ferulic acid (99%, Sigma Aldrich, St. Louis, MO, USA), chlorogenic acid (95%, Sigma Aldrich, St. Louis, MO, USA), syringic acid (95%, Sigma Aldrich, St. Louis, MO, USA), ellagic acid (95%, Sigma Aldrich, St. Louis, MO, USA), caffeic acid (95%, Sigma Aldrich, St. Louis, MO, USA), 2-hydroxycinnamic acid (97%, Sigma Aldrich, St. Louis, MO, USA), 3,4-dihydroxybenzoic acid (97%, Sigma Aldrich, St. Louis, MO, USA), trans-Sinapic acid (99%, Sigma Aldrich, St. Louis, MO, USA), trans-p-Coumaric acid (99.7%, Sigma Aldrich, St. Louis, MO, USA) and bidistilled water.

### 4.3. Sample Preparation

#### 4.3.1. Sample Handling

Different plant parts were divided into 10 groups: fresh samples (Group 1), samples that were dried at room temperature (Group 2), samples that were frozen in −18 °C (Group 3), −80 °C (Group 4), and −196 °C (Group 5) for 3 months, samples that were frozen in −18 °C (Group 6), −80 °C (Group 7), and −196 °C (Group 8) for 6 months, samples that were used for aerobic (Group 9) and anaerobic (Group 10) fermentation for 24, 48, and 72 h. Afterwards, fermented plant parts were dried at 35 °C and 60 °C.

Group No. 1 samples, after separation into different plant parts, were immediately cut into 3 ± 1 mm particles and used further for extraction of biologically active compounds. Group No. 2 samples were left to dry in a ventilated dark room for 7 days. After drying at room temperature, the samples were cut into 3 ± 1 mm particles for further analysis. Group No. 3–8 samples were frozen at respective temperatures for 3 and 6 months. After freezing, all samples were cut into particles of the same size as described previously. Fresh samples of Group No. 9 and 10 were compressed and crushed with hands to damage cell walls and release the inner fluids to start the natural fermentation process. Different plant part samples were put into plastic bags with access to atmospheric air for aerobic fermentation. All samples were put into plastic bags and sealed using a vacuum pump for anaerobic fermentation, creating anaerobic conditions.

After 24, 48, and 72 h natural aerobic and anaerobic fermentation processes, all different samples were cut into particles of the same size as described previously.

The moisture content in *C. angustifolium* samples was determined by PMB-53 Moisture Balance (Adam Equipment, Kingston, UK) according to the manufacturer’s recommendations. The moisture content in fresh and frozen leaf samples ranged from 70.1 to 81.9%, blossom samples—72.3 to 83.4%, and stem samples—62.8 to 69.4%. The moisture content in air-dried and fermented leave samples ranged from 3.87 to 6.79%, blossom samples—5.49 to 7.28%, and stem samples—8.41 to 12.09%.

#### 4.3.2. Direct Extraction

For the extraction of biologically active compounds, weighed plant samples (0.5 g) were diluted with aqueous methanol (0%, 25%, 50%, and 75% (*v*/*v*)) solution (20 mL), shaking the mixture in an orbital shaker for 24 h at 200 rpm. Subsequently, the mixtures were filtered through a paper filter and a membrane filter (0.45 µm) and prepared for spectrometric analysis. In total, 240 extracts were prepared.

### 4.4. Spectrometric Analysis

Total phenolic and flavonoid contents, and free radical scavenging assays were carried out using a UV-VIS spectrometer Milton Roy Spectronic (Ivyland, PA, USA). All measurements were compared to a standard curve prepared using rutin solution and expressed as rutin equivalents (mean) in mg of rutin standard per 1 g of dry raw material: RE mg/g. All measurements were performed in triplicate.

The total amount of phenolic compounds in plant extracts was determined by the Folin–Ciocalteu method, and the total content of flavonoids was determined by AlCl_3_ colourimetric method. Both methods are well-established and were used in our previous studies [16]. For the determination of phenolic compounds, the absorbance of the sample was measured at 760 nm, and for the total content of flavonoids, 407 nm absorbance was used. Antioxidant activity was carried out using a slightly modified DPPH radical scavenging method. A DPPH solution was prepared by dissolving 8 mg of DPPH in 100 mL of acetonitrile, 100 mL of methanol, and 200 mL of prepared acetate buffer (pH 5.5). A total of 3 mL of radical solution were mixed with 0.077 mL of sample extract solution. Spectrometric analysis was performed in the dark at 515 nm at room temperature with a 15 min reaction incubation.

### 4.5. High-Performance Liquid Chromatography Using Electrochemical Detection

Fresh *C. angustifolium* samples were not analysed in this study, as the purpose of HPLC-ED analysis was not to compare or identify degradation between fresh and air-dried samples but rather to identify which compounds remain detectable after the drying process. Blossoms and stems were not included in this study because spectrometric analysis indicated that most biologically active compounds were present only in leaves. Therefore, leaves were chosen as the sole focus for this analysis to ensure a more targeted and relevant investigation of the biologically active compounds.

Twelve phenolic compounds: gallic acid, vanillic acid, ferulic acid, chlorogenic acid, syringic acid, ellagic acid, caffeic acid, trans-sinapic acid, trans-p-coumaric acid, 2-hydroxycinnamic acid, 3,4-dihydroxybenzoic (protocatechuic) acid, and rutin, were used as external standards. Each standard was injected separately into the HPLC-ED system for the identification and quantification of the compounds. The retention times of the standards were obtained and compared with the chromatograms of the plant extracts for precise identification. Each identified compound concentration, expressed in rutin equivalents (RE) mg/g, was calculated using the rutin–calibration curve.

A gradient high-performance liquid chromatography (HPLC) system (ESA, Chelmsford, MA, USA) was implemented for the separation and electrochemical detection of extracted compounds. A 5 µL sample was manually injected using a 10 µL Hamilton^®^ syringe (model 705N). The system uses two ESA 582 model HPLC pumps to generate a high-pressure gradient. Separations were performed using a reversed-phase C18 column (80 × 4.6 mm, 3 µm particle size, 120 Å pore size). Detection was achieved with a model 5600 CoulArray electrochemical detector, equipped with an array of cells with potentials of 300, 500, and 700 mV. The mobile phase flow rate was 0.25 mL/min. Elution was performed using a two-component mobile phase. Solvent A consisted of 50 mM sodium dihydrogen phosphate (pH 3) with a 1% (vol) methanol additive, while Solvent B was a mixture of 100 mM sodium dihydrogen phosphate (pH 3), acetonitrile, and methanol (30:60:10 *v*/*v*). Antioxidants were eluted using the following gradient: 3% of B at 0 min, 100% of B at 46 min, 3% of B at 57 min, and 3% of B at 63 min, maintaining the flow rate of 0.25 mL/min throughout the analysis.

### 4.6. Statistical Analysis

The experimental data were analysed using SPSS Statistics for Windows, version 30.0.0 (SPSS Inc., Chicago, IL, USA). All assays were conducted in triplicate (*n* = 3), with results expressed as mean values ± standard deviation. To assess the significance of treatment effects on the response variables, the ANOVA test followed by Tukey’s Honestly Significant Difference (HSD) test was conducted. This approach allowed us to determine statistically significant differences between groups and validate the impact of each treatment. Calibration graphs were constructed using rutin standard solutions.

## 5. Conclusions

The study results clearly demonstrate that air-drying *C. angustifolium* leaves, blossoms, and stems rapidly decrease biologically active compounds. Both natural aerobic and anaerobic fermentation had a positive effect, although natural aerobic fermentation had a bigger influence on increasing the total content of phenolic compounds, flavonoids, and radical scavenging activity. Storing all the samples in liquid nitrogen minimally decreases the number of biologically active compounds. Consequently, the study results highlight the importance of selecting the appropriate combination of fermentation and extraction methods to maximize the efficiency of extracting biologically active compounds. Choosing the correct freezing temperature is also significant to maximizing the long-term stability of polyphenols and flavonoids and radical scavenging activity in plant raw materials.

## Figures and Tables

**Table 1 molecules-29-05530-t001:** Spectrometric analysis results of Group 1 samples expressed in RE mg/g, *n* = 3, *p* < 0.001.

Sample and Extraction Solvent	Total Content of Phenolic Compounds	Total Content of Flavonoids	Radical Scavenging Activity
Fresh leaves, H_2_O	178.58 ± 7.68	18.28 ± 0.71	149.84 ± 6.87
Fresh leaves, 25% (*v*/*v*) aqueous MeOH	239.86 ± 9.84	29.95 ± 1.21	221.54 ± 9.45
Fresh leaves, 50% (*v*/*v*) aqueous MeOH	341.11 ± 10.48	37.26 ± 1.46	265.66 ± 12.51
Fresh leaves, 75% (*v*/*v*) aqueous MeOH	396.82 ± 15.41	47.29 ± 2.08	359.56 ± 12.99
Fresh blossoms, H_2_O	128.07 ± 5.82	16.20 ± 0.45	113.47 ± 4.71
Fresh blossoms, 25% (*v*/*v*) aqueous MeOH	226.85 ± 8.41	40.75 ± 1.48	176.80 ± 7.56
Fresh blossoms, 50% (*v*/*v*) aqueous MeOH	320.47 ± 11.26	52.55 ± 2.09	227.10 ± 13.82
Fresh blossoms, 75% (*v*/*v*) aqueous MeOH	338.80 ± 15.69	57.19 ± 2.44	297.59 ± 14.62
Fresh stems, H_2_O	22.52 ± 1.01	3.22 ± 0.09	15.28 ± 0.52
Fresh stems, 25% (*v*/*v*) aqueous MeOH	26.07 ± 1.21	3.69 ± 0.11	21.69 ± 0.98
Fresh stems, 50% (*v*/*v*) aqueous MeOH	29.32 ± 1.35	4.07 ± 0.14	24.09 ± 1.12
Fresh stems, 75% (*v*/*v*) aqueous MeOH	33.90 ± 1.58	5.63 ± 0.22	27.43 ± 1.28

**Table 2 molecules-29-05530-t002:** Spectrometric analysis results of Group 2 samples expressed in RE mg/g, *n* = 3, *p* < 0.001.

Sample and Extraction Solvent	Total Content of Phenolic Compounds	Total Content of Flavonoids	Radical Scavenging Activity
Dried leaves, H_2_O	57.36 ± 1.79	10.74 ± 0.38	49.54 ± 1.75
Dried leaves, 25% (*v*/*v*) aqueous MeOH	65.10 ± 2.54	12.25 ± 0.41	54.24 ± 1.89
Dried leaves, 50% (*v*/*v*) aqueous MeOH	92.41 ± 3.78	16.78 ± 0.58	81.78 ± 2.45
Dried leaves, 75% (*v*/*v*) aqueous MeOH	105.22 ± 4.02	17.55 ± 0.59	86.84 ± 2.86
Dried blossoms, H_2_O	35.82 ± 1.44	7.39 ± 0.24	21.57 ± 0.89
Dried blossoms, 25% (*v*/*v*) aqueous MeOH	48.27 ± 1.67	16.57 ± 0.42	31.81 ± 0.98
Dried blossoms, 50% (*v*/*v*) aqueous MeOH	72.16 ± 2.96	20.13 ± 0.58	50.30 ± 2.05
Dried blossoms, 75% (*v*/*v*) aqueous MeOH	85.34 ± 3.11	22.64 ± 0.69	63.93 ± 2.79
Dried stems, H_2_O	11.18 ± 0.34	2.22 ± 0.04	7.58 ± 0.19
Dried stems, 25% (*v*/*v*) aqueous MeOH	11.57 ± 0.43	2.41 ± 0.08	8.39 ± 0.26
Dried stems, 50% (*v*/*v*) aqueous MeOH	12.77 ± 0.44	3.45 ± 0.14	9.21 ± 0.34
Dried stems, 75% (*v*/*v*) aqueous MeOH	14.41 ± 0.34	4.47 ± 0.18	11.11 ± 0.49

**Table 3 molecules-29-05530-t003:** Spectrometric analysis results of Group 9 samples expressed in RE mg/g, *n* = 3, *p* ≤ 0.05.

Sample, Aerobic Fermentation Time, Drying Temperature and Extraction Solvent	Total Content of Phenolic Compounds	Total Content of Flavonoids	Radical Scavenging Activity
Leaves, 24 h, 35 °C, H_2_O	98.27 ± 3.52	22.88 ± 0.81	77.49 ± 2.74
Leaves, 24 h, 35 °C, 25% (*v*/*v*) aqueous MeOH	117.48 ± 4.87	27.90 ± 0.96	97.72 ± 3.42
Leaves, 24 h, 35 °C, 50% (*v*/*v*) aqueous MeOH	146.73 ± 5.66	28.31 ± 1.21	128.65 ± 5.41
Leaves, 24 h, 35 °C, 75% (*v*/*v*) aqueous MeOH	168.06 ± 7.96	30.13 ± 1.31	153.40 ± 6.23
Leaves, 24 h, 60 °C, H_2_O	64.73 ± 2.41	19.06 ± 0.74	48.01 ± 1.48
Leaves, 24 h, 60 °C, 25% (*v*/*v*) aqueous MeOH	83.52 ± 3.64	20.72 ± 0.82	67.51 ± 2.31
Leaves, 24 h, 60 °C, 50% (*v*/*v*) aqueous MeOH	99.05 ± 4.22	22.19 ± 0.96	89.36 ± 3.48
Leaves, 24 h, 60 °C, 75% (*v*/*v*) aqueous MeOH	121.59 ± 4.69	23.89 ± 0.99	118.16 ± 4.19
Leaves, 48 h, 35 °C, H_2_O	108.12 ± 3.41	24.71 ± 0.69	87.25 ± 3.24
Leaves, 48 h, 35 °C, 25% (*v*/*v*) aqueous MeOH	129.99 ± 5.94	30.30 ± 0.41	109.61 ± 4.18
Leaves, 48 h, 35 °C, 50% (*v*/*v*) aqueous MeOH	145.28 ± 5.26	33.03 ± 0.62	139.36 ± 2.91
Leaves, 48 h, 35 °C, 75% (*v*/*v*) aqueous MeOH	172.95 ± 4.69	36.53 ± 0.83	160.06 ± 4.69
Leaves, 48 h, 60 °C, H_2_O	72.29 ± 2.17	18.31 ± 0.71	55.96 ± 1.95
Leaves, 48 h, 60 °C, 25% (*v*/*v*) aqueous MeOH	94.04 ± 3.41	21.79 ± 0.52	79.16 ± 2.36
Leaves, 48 h, 60 °C, 50% (*v*/*v*) aqueous MeOH	108.54 ± 4.66	23.44 ± 0.67	105.27 ± 4.16
Leaves, 48 h, 60 °C, 75% (*v*/*v*) aqueous MeOH	130.41 ± 5.18	24.87 ± 0.39	131.72 ± 5.91
Leaves, 72 h, 35 °C, H_2_O	79.84 ± 2.84	16.20 ± 0.41	64.41 ± 2.94
Leaves, 72 h, 35 °C, 25% (*v*/*v*) aqueous MeOH	104.31 ± 4.69	23.43 ± 0.69	90.22 ± 3.41
Leaves, 72 h, 35 °C, 50% (*v*/*v*) aqueous MeOH	128.30 ± 5.41	26.11 ± 0.89	114.97 ± 4.91
Leaves, 72 h, 35 °C, 75% (*v*/*v*) aqueous MeOH	152.77 ± 5.62	28.49 ± 1.21	138.64 ± 5.97
Leaves, 72 h, 60 °C, H_2_O	53.92 ± 1.18	15.43 ± 0.64	40.39 ± 1.26
Leaves, 72 h, 60 °C, 25% (*v*/*v*) aqueous MeOH	73.43 ± 2.69	17.88 ± 0.41	53.61 ± 1.39
Leaves, 72 h, 60 °C, 50% (*v*/*v*) aqueous MeOH	98.99 ± 3.41	19.60 ± 0.36	71.88 ± 2.58
Leaves, 72 h, 60 °C, 75% (*v*/*v*) aqueous MeOH	115.01 ± 4.96	21.70 ± 0.89	95.52 ± 3.94
Blossoms, 24 h, 35 °C, H_2_O	57.00 ± 1.74	28.61 ± 1.22	51.10 ± 2.36
Blossoms, 24 h, 35 °C, 25% (*v*/*v*) aqueous MeOH	85.70 ± 2.69	30.15 ± 1.34	68.64 ± 2.48
Blossoms, 24 h, 35 °C, 50% (*v*/*v*) aqueous MeOH	108.32 ± 3.48	32,16 ± 1.39	85.69 ± 3.41
Blossoms, 24 h, 35 °C, 75% (*v*/*v*) aqueous MeOH	128.12 ± 5.21	36.18 ± 1.42	109.39 ± 4.96
Blossoms, 24 h, 60 °C, H_2_O	37.16 ± 1.16	19.01 ± 0.81	38.90 ± 1.28
Blossoms, 24 h, 60 °C, 25% (*v*/*v*) aqueous MeOH	50.50 ± 1.39	24.21 ± 0.98	53.83 ± 1.69
Blossoms, 24 h, 60 °C, 50% (*v*/*v*) aqueous MeOH	78.36 ± 2.48	26.42 ± 1.12	71.95 ± 2.36
Blossoms, 24 h, 60 °C, 75% (*v*/*v*) aqueous MeOH	99.36 ± 3.96	30.18 ± 1.26	87.83 ± 3.49
Blossoms, 48 h, 35 °C, H_2_O	65.60 ± 2.69	31.22 ± 1.24	62.59 ± 2.16
Blossoms, 48 h, 35 °C, 25% (*v*/*v*) aqueous MeOH	90.09 ± 2.33	32.42 ± 1.31	75.03 ± 2.69
Blossoms, 48 h, 35 °C, 50% (*v*/*v*) aqueous MeOH	95.87 ± 3.96	34.12 ± 1.40	105.24 ± 4.26
Blossoms, 48 h, 35 °C, 75% (*v*/*v*) aqueous MeOH	139.55 ± 5.91	37.98 ± 1.37	121.59 ± 4.91
Blossoms, 48 h, 60 °C, H_2_O	44.72 ± 1.37	22.10 ± 0.87	46.48 ± 1.23
Blossoms, 48 h, 60 °C, 25% (*v*/*v*) aqueous MeOH	64.22 ± 2.69	27.28 ± 0.69	60.94 ± 2.69
Blossoms, 48 h, 60 °C, 50% (*v*/*v*) aqueous MeOH	87.62 ± 4.87	29.82 ± 0.83	84.15 ± 2.42
Blossoms, 48 h, 60 °C, 75% (*v*/*v*) aqueous MeOH	109.83 ± 4.69	33.16 ± 0.82	106.66 ± 3.19
Blossoms, 72 h, 35 °C, H_2_O	45.93 ± 1.19	20.32 ± 0.49	42.93 ± 1.71
Blossoms, 72 h, 35 °C, 25% (*v*/*v*) aqueous MeOH	76.25 ± 2.64	23.02 ± 0.97	52.41 ± 2.14
Blossoms, 72 h, 35 °C, 50% (*v*/*v*) aqueous MeOH	96.23 ± 4.29	30.66 ± 0.67	71.12 ± 2.69
Blossoms, 72 h, 35 °C, 75% (*v*/*v*) aqueous MeOH	114.52 ± 4.93	33.04 ± 1.24	94.58 ± 3.48
Blossoms, 72 h, 60 °C, H_2_O	28.78 ± 1.27	14.77 ± 0.64	30.37 ± 1.11
Blossoms, 72 h, 60 °C, 25% (*v*/*v*) aqueous MeOH	43.88 ± 1.96	20.36 ± 0.69	43.64 ± 1.27
Blossoms, 72 h, 60 °C, 50% (*v*/*v*) aqueous MeOH	68.97 ± 2.49	22.52 ± 0.72	51.34 ± 2.43
Blossoms, 72 h, 60 °C, 75% (*v*/*v*) aqueous MeOH	86.42 ± 3.69	24.55 ± 0.69	68.40 ± 3.11
Stems, 24 h, 35 °C, H_2_O	10.37 ± 0.39	4.23 ± 0.41	8.70 ± 0.12
Stems, 24 h, 35 °C, 25% (*v*/*v*) aqueous MeOH	13.80 ± 0.49	4.84 ± 0.69	11.73 ± 0.34
Stems, 24 h, 35 °C, 50% (*v*/*v*) aqueous MeOH	20.72 ± 0.69	5.58 ± 0.89	13.57 ± 0.41
Stems, 24 h, 35 °C, 75% (*v*/*v*) aqueous MeOH	26.86 ± 0.99	6.48 ± 0.39	17.14 ± 0.68
Stems, 24 h, 60 °C, H_2_O	6.68 ± 0.21	3.34 ± 0.06	6.86 ± 0.24
Stems, 24 h, 60 °C, 25% (*v*/*v*) aqueous MeOH	10.24 ± 0.49	4.24 ± 0.15	9.15 ± 0.34
Stems, 24 h, 60 °C, 50% (*v*/*v*) aqueous MeOH	16.51 ± 0.39	4.95 ± 0.16	11.70 ± 0.41
Stems, 24 h, 60 °C, 75% (*v*/*v*) aqueous MeOH	23.56 ± 0.79	5.99 ± 0.21	13.54 ± 0.42
Stems, 48 h, 35 °C, H_2_O	13.67 ± 0.55	5.39 ± 0.09	10.07 ± 0.46
Stems, 48 h, 35 °C, 25% (*v*/*v*) aqueous MeOH	19.42 ± 0.69	6.70 ± 0.15	13.57 ± 0.26
Stems, 48 h, 35 °C, 50% (*v*/*v*) aqueous MeOH	28.41 ± 0.95	8.38 ± 0.16	16.57 ± 0.63
Stems, 48 h, 35 °C, 75% (*v*/*v*) aqueous MeOH	33.07 ± 1.12	9.35 ± 0.19	20.90 ± 0.91
Stems, 48 h, 60 °C, H_2_O	10.82 ± 0.26	3.98 ± 0.22	8.74 ± 0.23
Stems, 48 h, 60 °C, 25% (*v*/*v*) aqueous MeOH	13.28 ± 0.39	4.75 ± 0.31	12.33 ± 0.13
Stems, 48 h, 60 °C, 50% (*v*/*v*) aqueous MeOH	18.58 ± 0.69	5.33 ± 0.24	13.61 ± 0.45
Stems, 48 h, 60 °C, 75% (*v*/*v*) aqueous MeOH	28.29 ± 1.29	6.00 ± 0.26	15.84 ± 0.61
Stems, 72 h, 35 °C, H_2_O	7.26 ± 0.21	3.92 ± 0.14	6.67 ± 0.23
Stems, 72 h, 35 °C, 25% (*v*/*v*) aqueous MeOH	9.20 ± 0.42	4.43 ± 0.12	9.72 ± 0.19
Stems, 72 h, 35 °C, 50% (*v*/*v*) aqueous MeOH	11.86 ± 0.36	5.07 ± 0.19	12.43 ± 0.36
Stems, 72 h, 35 °C, 75% (*v*/*v*) aqueous MeOH	20.91 ± 0.87	5.58 ± 0.22	13.96 ± 0.39
Stems, 72 h, 60 °C, H_2_O	3.38 ± 0.11	2.77 ± 0.11	5.84 ± 0.21
Stems, 72 h, 60 °C, 25% (*v*/*v*) aqueous MeOH	7.59 ± 0.23	3.41 ± 0.13	8.23 ± 0.36
Stems, 72 h, 60 °C, 50% (*v*/*v*) aqueous MeOH	11.40 ± 0.41	4.40 ± 0.14	9.69 ± 0.27
Stems, 72 h, 60 °C, 75% (*v*/*v*) aqueous MeOH	19.23 ± 0.86	4.69 ± 0.16	11.15 ± 0.46

**Table 4 molecules-29-05530-t004:** Spectrometric analysis results of Group 10 samples expressed in RE mg/g, *n* = 3, *p* ≤ 0.05.

Sample, Anaerobic Fermentation Time, Drying Temperature, Extraction Solvent	Total Content of Phenolic Compounds	Total Content of Flavonoids	Radical Scavenging Activity
Leaves, 24 h, 35 °C, H_2_O	84.91 ± 3.12	19.60 ± 0.41	69.05 ± 2.41
Leaves, 24 h, 35 °C, 25% (*v*/*v*) aqueous MeOH	105.34 ± 4.19	22.67 ± 0.58	91.17 ± 3.64
Leaves, 24 h, 35 °C, 50% (*v*/*v*) aqueous MeOH	132.47 ± 5.27	26.71 ± 0.69	121.51 ± 5.12
Leaves, 24 h, 35 °C, 75% (*v*/*v*) aqueous MeOH	142.50 ± 5.34	28.49 ± 0.82	133.76 ± 6.44
Leaves, 24 h, 60 °C, H_2_O	46.67 ± 1.26	14.67 ± 0.51	35.46 ± 0.95
Leaves, 24 h, 60 °C, 25% (*v*/*v*) aqueous MeOH	74.10 ± 2.41	18.01 ± 0.71	46.44 ± 1.24
Leaves, 24 h, 60 °C, 50% (*v*/*v*) aqueous MeOH	90.35 ± 3.44	19.82 ± 0.64	64.03 ± 2.03
Leaves, 24 h, 60 °C, 75% (*v*/*v*) aqueous MeOH	109.99 ± 3.68	22.05 ± 0.86	87.57 ± 3.65
Leaves, 48 h, 35 °C, H_2_O	91.08 ± 3.59	21.26 ± 0.89	78.44 ± 2.95
Leaves, 48 h, 35 °C, 25% (*v*/*v*) aqueous MeOH	119.05 ± 4.19	25.20 ± 0.64	102.60 ± 4.23
Leaves, 48 h, 35 °C, 50% (*v*/*v*) aqueous MeOH	135.73 ± 5.83	27.94 ± 0.33	133.88 ± 5.21
Leaves, 48 h, 35 °C, 75% (*v*/*v*) aqueous MeOH	151.92 ± 5.34	30.27 ± 1.02	142.21 ± 5.81
Leaves, 48 h, 60 °C, H_2_O	60.32 ± 2.69	16.34 ± 0.55	45.99 ± 2.03
Leaves, 48 h, 60 °C, 25% (*v*/*v*) aqueous MeOH	77.84 ± 2.48	19.58 ± 0.58	57.20 ± 2.41
Leaves, 48 h, 60 °C, 50% (*v*/*v*) aqueous MeOH	98.99 ± 3.67	21.20 ± 0.66	71.65 ± 2.66
Leaves, 48 h, 60 °C, 75% (*v*/*v*) aqueous MeOH	121.21 ± 5.94	23.85 ± 0.89	95.08 ± 3.09
Leaves, 72 h, 35 °C, H_2_O	68.90 ± 2.49	13.81 ± 0.51	58.70 ± 1.84
Leaves, 72 h, 35 °C, 25% (*v*/*v*) aqueous MeOH	94.76 ± 3.64	17.44 ± 0.75	77.73 ± 3.41
Leaves, 72 h, 35 °C, 50% (*v*/*v*) aqueous MeOH	110.84 ± 4.96	22.69 ± 0.67	107.47 ± 4.69
Leaves, 72 h, 35 °C, 75% (*v*/*v*) aqueous MeOH	131.86 ± 5.61	24.66 ± 0.39	128.65 ± 5.41
Leaves, 72 h, 60 °C, H_2_O	43.10 ± 1.37	9.59 ± 0.41	27.72 ± 1.02
Leaves, 72 h, 60 °C, 25% (*v*/*v*) aqueous MeOH	65.10 ± 2.69	16.71 ± 0.75	34.45 ± 1.22
Leaves, 72 h, 60 °C, 50% (*v*/*v*) aqueous MeOH	83.22 ± 3.41	18.48 ± 0.82	48.34 ± 1.65
Leaves, 72 h, 60 °C, 75% (*v*/*v*) aqueous MeOH	101.29 ± 4.66	19.70 ± 0.46	69.19 ± 2.33
Blossoms, 24 h, 35 °C, H_2_O	48.39 ± 1.36	21.13 ± 0.89	42.69 ± 1.25
Blossoms, 24 h, 35 °C, 25% (*v*/*v*) aqueous MeOH	67.77 ± 2.44	23.00 ± 0.15	55.84 ± 2.09
Blossoms, 24 h, 35 °C, 50% (*v*/*v*) aqueous MeOH	88.95 ± 3.22	25.94 ± 0.89	75.03 ± 2.41
Blossoms, 24 h, 35 °C, 75% (*v*/*v*) aqueous MeOH	105.79 ± 4.33	29.04 ± 1.13	93.51 ± 3.64
Blossoms, 24 h, 60 °C, H_2_O	30.34 ± 0.89	14.52 ± 0.65	27.88 ± 1.09
Blossoms, 24 h, 60 °C, 25% (*v*/*v*) aqueous MeOH	43.04 ± 1.64	19.32 ± 0.89	41.39 ± 1.77
Blossoms, 24 h, 60 °C, 50% (*v*/*v*) aqueous MeOH	68.19 ± 2.49	23.61 ± 0.87	51.81 ± 2.15
Blossoms, 24 h, 60 °C, 75% (*v*/*v*) aqueous MeOH	84.07 ± 3.66	26.78 ± 1.15	69.35 ± 3.41
Blossoms, 48 h, 35 °C, H_2_O	51.10 ± 1.48	23.04 ± 0.89	51.34 ± 1.55
Blossoms, 48 h, 35 °C, 25% (*v*/*v*) aqueous MeOH	80.16 ± 3.64	25.96 ± 0.88	68.87 ± 3.14
Blossoms, 48 h, 35 °C, 50% (*v*/*v*) aqueous MeOH	97.43 ± 3.11	28.56 ± 1.24	85.46 ± 4.06
Blossoms, 48 h, 35 °C, 75% (*v*/*v*) aqueous MeOH	112.89 ± 4.66	32.84 ± 1.29	96.59 ± 4.56
Blossoms, 48 h, 60 °C, H_2_O	40.51 ± 1.66	16.20 ± 0.72	37.60 ± 0.99
Blossoms, 48 h, 60 °C, 25% (*v*/*v*) aqueous MeOH	48.94 ± 2.11	20.81 ± 0.69	50.39 ± 1.58
Blossoms, 48 h, 60 °C, 50% (*v*/*v*) aqueous MeOH	78.72 ± 3.09	25.68 ± 1.01	64.61 ± 2.34
Blossoms, 48 h, 60 °C, 75% (*v*/*v*) aqueous MeOH	94.84 ± 3.46	28.46 ± 1.21	81.43 ± 3.44
Blossoms, 72 h, 35 °C, H_2_O	39.19 ± 1.16	16.61 ± 0.78	35.94 ± 0.87
Blossoms, 72 h, 35 °C, 25% (*v*/*v*) aqueous MeOH	57.96 ± 1.26	19.40 ± 0.55	51.22 ± 1.65
Blossoms, 72 h, 35 °C, 50% (*v*/*v*) aqueous MeOH	82.57 ± 2.44	21.80 ± 1.02	66.74 ± 2.64
Blossoms, 72 h, 35 °C, 75% (*v*/*v*) aqueous MeOH	95.21 ± 3.45	24.24 ± 0.66	86.52 ± 3.44
Blossoms, 72 h, 60 °C, H_2_O	24.45 ± 0.64	11.56 ± 0.45	21.49 ± 1.07
Blossoms, 72 h, 60 °C, 25% (*v*/*v*) aqueous MeOH	31.01 ± 1.31	16.20 ± 0.80	24.40 ± 1.65
Blossoms, 72 h, 60 °C, 50% (*v*/*v*) aqueous MeOH	60.49 ± 2.67	21.09 ± 0.56	50.39 ± 1.36
Blossoms, 72 h, 60 °C, 75% (*v*/*v*) aqueous MeOH	72.10 ± 2.33	23.45 ± 1.03	56.20 ± 1.87
Stems, 24 h, 35 °C, H_2_O	7.46 ± 0.22	2.87 ± 0.05	6.76 ± 0.22
Stems, 24 h, 35 °C, 25% (*v*/*v*) aqueous MeOH	10.95 ± 0.42	3.37 ± 0.10	9.21 ± 0.34
Stems, 24 h, 35 °C, 50% (*v*/*v*) aqueous MeOH	15.03 ± 0.64	4.49 ± 0.18	11.92 ± 0.51
Stems, 24 h, 35 °C, 75% (*v*/*v*) aqueous MeOH	22.66 ± 0.69	5.39 ± 0.22	14.21 ± 0.65
Stems, 24 h, 60 °C, H_2_O	5.19 ± 0.16	2.48 ± 0.06	4.94 ± 0.21
Stems, 24 h, 60 °C, 25% (*v*/*v*) aqueous MeOH	8.43 ± 0.34	2.99 ± 0.06	8.07 ± 0.34
Stems, 24 h, 60 °C, 50% (*v*/*v*) aqueous MeOH	12.31 ± 0.55	3.66 ± 0.07	10.17 ± 0.25
Stems, 24 h, 60 °C, 75% (*v*/*v*) aqueous MeOH	19.10 ± 0.86	4.15 ± 0.09	11.35 ± 0.41
Stems, 48 h, 35 °C, H_2_O	8.56 ± 0.12	3.47 ± 0.15	7.88 ± 0.22
Stems, 48 h, 35 °C, 25% (*v*/*v*) aqueous MeOH	18.07 ± 0.33	4.33 ± 0.21	10.80 ± 0.41
Stems, 48 h, 35 °C, 50% (*v*/*v*) aqueous MeOH	23.89 ± 0.64	5.38 ± 0.19	13.48 ± 0.52
Stems, 48 h, 35 °C, 75% (*v*/*v*) aqueous MeOH	30.55 ± 1.22	6.31 ± 0.21	16.15 ± 0.29
Stems, 48 h, 60 °C, H_2_O	6.81 ± 0.22	3.12 ± 0.05	7.30 ± 0.21
Stems, 48 h, 60 °C, 25% (*v*/*v*) aqueous MeOH	15.93 ± 0.64	3.95 ± 0.10	8.64 ± 0.33
Stems, 48 h, 60 °C, 50% (*v*/*v*) aqueous MeOH	21.88 ± 0.98	4.55 ± 0.14	10.65 ± 0.41
Stems, 48 h, 60 °C, 75% (*v*/*v*) aqueous MeOH	26.99 ± 1.31	4.98 ± 0.21	11.76 ± 0.52
Stems, 72 h, 35 °C, H_2_O	5.71 ± 0.17	2.09 ± 0.06	5.52 ± 0.18
Stems, 72 h, 35 °C, 25% (*v*/*v*) aqueous MeOH	9.27 ± 0.31	3.24 ± 0.11	7.68 ± 0.24
Stems, 72 h, 35 °C, 50% (*v*/*v*) aqueous MeOH	13.47 ± 0.51	3.72 ± 0.18	9.80 ± 0.46
Stems, 72 h, 35 °C, 75% (*v*/*v*) aqueous MeOH	18.84 ± 0.34	4.37 ± 0.05	11.18 ± 0.42
Stems, 72 h, 60 °C, H_2_O	2.74 ± 0.11	1.78 ± 0.08	1.96 ± 0.10
Stems, 72 h, 60 °C, 25% (*v*/*v*) aqueous MeOH	7.01 ± 0.34	2.52 ± 0.05	5.10 ± 0.25
Stems, 72 h, 60 °C, 50% (*v*/*v*) aqueous MeOH	9.72 ± 0.48	3.06 ± 0.06	7.24 ± 0.33
Stems, 72 h, 60 °C, 75% (*v*/*v*) aqueous MeOH	16.64 ± 0.64	3.51 ± 0.07	10.57 ± 0.41

**Table 5 molecules-29-05530-t005:** Spectrometric analysis results of Groups 3–5 samples expressed in RE mg/g, *n* = 3, *p* ≤ 0.05.

Sample, Freezing Temperature, Extraction Solvent	Total Content of Phenolic Compounds	Total Content of Flavonoids	Radical Scavenging Activity
Leaves, −18 °C, H_2_O	128.48 ± 2.42	13.16 ± 0.43	110.89 ± 3.34
Leaves, −18 °C, 25% (*v*/*v*) aqueous MeOH	174.75 ± 3.89	23.16 ± 1.11	167.18 ± 3.24
Leaves, −18 °C, 50% (*v*/*v*) aqueous MeOH	255.06 ± 4.82	27.32 ± 1.23	200.42 ± 4.51
Leaves, −18 °C, 75% (*v*/*v*) aqueous MeOH	306.93 ± 7.12	31.77 ± 1.84	207.41 ± 5.54
Blossoms, −18 °C, H_2_O	91.88 ± 3.48	11.70 ± 0.68	85.14 ± 3.26
Blossoms, −18 °C, 25% (*v*/*v*) aqueous MeOH	164.98 ± 4.55	31.58 ± 1.14	131.75 ± 4.65
Blossoms, −18 °C, 50% (*v*/*v*) aqueous MeOH	220.32 ± 5.09	38.01 ± 1.58	217.00 ± 5.99
Blossoms, −18 °C, 75% (*v*/*v*) aqueous MeOH	247.66 ± 6.07	39.96 ± 1.74	230.10 ± 6.18
Stems, −18 °C, H_2_O	17.46 ± 0.52	2.17 ± 0.07	11.32 ± 0.41
Stems, −18 °C, 25% (*v*/*v*) aqueous MeOH	19.87 ± 0.87	2.51 ± 0.09	16.04 ± 0.58
Stems, −18 °C, 50% (*v*/*v*) aqueous MeOH	22.72 ± 0.94	2.85 ± 0.14	17.23 ± 0.62
Stems, −18 °C, 75% (*v*/*v*) aqueous MeOH	25.73 ± 1.25	4.03 ± 0.21	20.91 ± 1.00
Leaves, −80 °C, H_2_O extraction	158.21 ± 2.91	15.55 ± 0.64	126.53 ± 5.88
Leaves, −80 °C, 25% (*v*/*v*) aqueous MeOH	214.89 ± 4.41	24.60 ± 0.99	179.34 ± 5.41
Leaves, −80 °C, 50% (*v*/*v*) aqueous MeOH	287.41 ± 5.63	30.66 ± 1.18	229.29 ± 6.54
Leaves, −80 °C, 75% (*v*/*v*) aqueous MeOH	346.06 ± 7.88	40.77 ± 1.89	243.37 ± 7.88
Blossoms, −80 °C, H_2_O	109.02 ± 3.41	13.98 ± 0.58	92.36 ± 3.21
Blossoms, −80 °C, 25% (*v*/*v*) aqueous MeOH	189.79 ± 4.69	36.21 ± 1.19	143.95 ± 4.78
Blossoms, −80 °C, 50% (*v*/*v*) aqueous MeOH	281.82 ± 5.91	45.74 ± 2.01	239.49 ± 5.55
Blossoms, −80 °C, 75% (*v*/*v*) aqueous MeOH	300.48 ± 6.79	49.58 ± 3.05	259.64 ± 6.19
Stems, −80 °C, H_2_O	19.48 ± 0.67	2.79 ± 0.11	12.47 ± 0.54
Stems, −80 °C, 25% (*v*/*v*) aqueous MeOH	22.34 ± 0.89	3.06 ± 0.12	18.70 ± 0.58
Stems, −80 °C, 50% (*v*/*v*) aqueous MeOH	25.98 ± 1.16	3.54 ± 0.16	20.02 ± 0.98
Stems, −80 °C, 75% (*v*/*v*) aqueous MeOH	30.01 ± 1.45	4.70 ± 0.22	22.99 ± 1.01
Leaves, −196 °C, H_2_O	166.71 ± 2.91	17.49 ± 0.77	143.46 ± 2.89
Leaves, −196 °C, 25% (*v*/*v*) aqueous MeOH	235.77 ± 2.33	29.77 ± 1.42	210.98 ± 4.96
Leaves, −196 °C, 50% (*v*/*v*) aqueous MeOH	334.96 ± 4.91	35.99 ± 1.18	254.35 ± 5.81
Leaves, −196 °C, 75% (*v*/*v*) aqueous MeOH	388.27 ± 3.93	43.15 ± 0.68	271.53 ± 4.82
Blossoms, −196 °C, H_2_O	120.67 ± 4.74	15.52 ± 0.63	111.56 ± 2.26
Blossoms, −196 °C, 25% (*v*/*v*) aqueous MeOH	218.38 ± 5.63	37.76 ± 1.08	165.27 ± 3.71
Blossoms, −196 °C, 50% (*v*/*v*) aqueous MeOH	312.76 ± 5.82	50.31 ± 1.41	279.08 ± 2.48
Blossoms, −196 °C, 75% (*v*/*v*) aqueous MeOH	328.78 ± 3.81	53.23 ± 0.44	289.74 ± 1.49
Stems, −196 °C, H_2_O	21.43 ± 0.18	3.12 ± 0.14	14.47 ± 0.67
Stems, −196 °C, 25% (*v*/*v*) aqueous MeOH	25.68 ± 0.19	3.43 ± 0.16	20.41 ± 0.88
Stems, −196 °C, 50% (*v*/*v*) aqueous MeOH	28.18 ± 0.16	4.00 ± 0.15	22.47 ± 1.05
Stems, −196 °C, 75% (*v*/*v*) aqueous MeOH	31.54 ± 0.25	5.32 ± 0.06	26.50 ± 0.77

**Table 6 molecules-29-05530-t006:** Spectrometric analysis results of Groups 6–8 samples expressed in RE mg/g, *n* = 3, *p* ≤ 0.05.

Sample, Freezing Temperature, Extraction Solvent	Total Content of Phenolic Compounds	Total Content of Flavonoids	Radical Scavenging Activity
Leaves, −18 °C, H_2_O	105.00 ± 2.29	9.88 ± 0.23	87.14 ± 2.59
Leaves, −18 °C, 25% (*v*/*v*) aqueous MeOH	145.29 ± 3.11	17.16 ± 0.46	129.48 ± 3.49
Leaves, −18 °C, 50% (*v*/*v*) aqueous MeOH	219.63 ± 4.94	22.49 ± 0.89	171.18 ± 3.91
Leaves, −18 °C, 75% (*v*/*v*) aqueous MeOH	280.32 ± 5.19	28.21 ± 0.99	202.30 ± 4.64
Blossoms, −18 °C, H_2_O	70.39 ± 2.64	11.02 ± 0.18	79.20 ± 2.48
Blossoms, −18 °C, 25% (*v*/*v*) aqueous MeOH	125.28 ± 3.69	24.42 ± 0.56	123.15 ± 3.73
Blossoms, −18 °C, 50% (*v*/*v*) aqueous MeOH	169.13 ± 4.19	31.97 ± 0.69	183.43 ± 3.37
Blossoms, −18 °C, 75% (*v*/*v*) aqueous MeOH	221.27 ± 4.58	40.07 ± 1.45	214.56 ± 4.63
Stems, −18 °C, H_2_O	15.72 ± 0.64	1.69 ± 0.04	9.83 ± 0.15
Stems, −18 °C, 25% (*v*/*v*) aqueous MeOH	16.85 ± 0.79	2.62 ± 0.09	13.32 ± 0.23
Stems, −18 °C, 50% (*v*/*v*) aqueous MeOH	21.18 ± 0.98	2.71 ± 0.15	17.50 ± 0.74
Stems, −18 °C, 75% (*v*/*v*) aqueous MeOH	22.64 ± 1.01	4.03 ± 0.51	18.36 ± 0.59
Leaves, −80 °C, H_2_O extraction	134.77 ± 2.89	13.21 ± 0.41	113.29 ± 2.99
Leaves, −80 °C, 25% (*v*/*v*) aqueous MeOH	177.03 ± 3.41	22.49 ± 0.59	160.56 ± 3.33
Leaves, −80 °C, 50% (*v*/*v*) aqueous MeOH	259.06 ± 4.83	29.16 ± 1.11	196.38 ± 4.91
Leaves, −80 °C, 75% (*v*/*v*) aqueous MeOH	319.92 ± 5.22	33.21 ± 1.45	217.03 ± 5.09
Blossoms, −80 °C, H_2_O	98.56 ± 1.02	12.82 ± 0.49	87.50 ± 1.91
Blossoms, −80 °C, 25% (*v*/*v*) aqueous MeOH	184.40 ± 2.09	31.22 ± 0.89	128.92 ± 3.67
Blossoms, −80 °C, 50% (*v*/*v*) aqueous MeOH	226.30 ± 4.55	41.02 ± 1.19	202.70 ± 4.91
Blossoms, −80 °C, 75% (*v*/*v*) aqueous MeOH	269.44 ± 4.96	42.95 ± 1.59	224.00 ± 5.66
Stems, −80 °C, H_2_O	16.68 ± 0.48	2.42 ± 0.08	11.79 ± 0.39
Stems, −80 °C, 25% (*v*/*v*) aqueous MeOH	20.76 ± 0.53	2.86 ± 0.10	16.55 ± 0.42
Stems, −80 °C, 50% (*v*/*v*) aqueous MeOH	23.81 ± 0.86	2.99 ± 0.12	19.08 ± 0.55
Stems, −80 °C, 75% (*v*/*v*) aqueous MeOH	27.02 ± 0.99	4.34 ± 0.19	21.41 ± 0.83
Leaves, −196 °C, H_2_O	159.16 ± 2.48	16.42 ± 0.67	132.62 ± 2.43
Leaves, −196 °C, 25% (*v*/*v*) aqueous MeOH	212.75 ± 4.43	26.60 ± 1.21	196.48 ± 3.79
Leaves, −196 °C, 50% (*v*/*v*) aqueous MeOH	302.75 ± 3.79	32.81 ± 1.41	238.85 ± 4.66
Leaves, −196 °C, 75% (*v*/*v*) aqueous MeOH	355.64 ± 5.49	41.93 ± 1.68	244.45 ± 5.73
Blossoms, −196 °C, H_2_O	110.96 ± 2.33	13.92 ± 0.57	99.18 ± 2.79
Blossoms, −196 °C, 25% (*v*/*v*) aqueous MeOH	202.86 ± 4.71	36.73 ± 1.43	155.45 ± 3.48
Blossoms, −196 °C, 50% (*v*/*v*) aqueous MeOH	283.64 ± 6.19	47.64 ± 2.08	251.82 ± 4.69
Blossoms, −196 °C, 75% (*v*/*v*) aqueous MeOH	302.68 ± 6.79	49.92 ± 2.12	275.54 ± 4.79
Stems, −196 °C, H_2_O	20.45 ± 0.87	2.90 ± 0.12	13.97 ± 0.58
Stems, −196 °C, 25% (*v*/*v*) aqueous MeOH	23.49 ± 0.99	3.24 ± 0.14	19.52 ± 0.78
Stems, −196 °C, 50% (*v*/*v*) aqueous MeOH	26.83 ± 1.21	3.71 ± 0.18	21.58 ± 1.01
Stems, −196 °C, 75% (*v*/*v*) aqueous MeOH	30.53 ± 1.02	5.03 ± 0.22	24.25 ± 1.08

**Table 7 molecules-29-05530-t007:** Identification and quantification of phenolic compounds in dried *C. angustifolium* leaf extracts, expressed in RE mg/g, *n* = 3, *p* ≤ 0.05.

Sample and Extraction Solvent	Compound	t_R_, min *	C_RE_, mg/g **
Dried leaves, H_2_O	Oenothein B	7.14 ± 0.05	6.01 ± 0.01
	Chlorogenic acid	30.59 ± 0.10	41.72 ± 0.08
	Rutin	40.05 ± 0.06	13.76 ± 0.01
	Ellagic acid	41.72 ± 0.09	<0.1 ± 0.01
Dried leaves, 25% (*v*/*v*) aqueous MeOH	Oenothein B	7.16 ± 0.04	19.28 ± 0.07
	Chlorogenic acid	30.69 ± 0.08	44.68 ± 0.06
	Rutin	40.02 ± 0.04	15.05 ± 0.02
	Ellagic acid	41.71 ± 0.06	<0.1 ± 0.01
Dried leaves, 50% (*v*/*v*) aqueous MeOH	Oenothein B	7.15 ± 0.06	31.85 ± 0.09
	Chlorogenic acid	30.43 ± 0.10	47.05 ± 0.07
	Rutin	40.03 ± 0.07	15.60 ± 0.02
	Ellagic acid	41.81 ± 0.06	<0.1 ± 0.01
Dried leaves, 75% (*v*/*v*) aqueous MeOH	Oenothein B	7.17 ± 0.02	64.36 ± 0.10
	Chlorogenic acid	30.50 ± 0.09	51.65 ± 0.09
	Rutin	40.10 ± 0.05	15.81 ± 0.02
	Ellagic acid	41.79 ± 0.08	<0.1 ± 0.01

* Retention time in min. ** Concentration expressed in Rutin Equivalents (RE) in mg/g.

**Table 8 molecules-29-05530-t008:** Identification and quantification of phenolic compounds in natural aerobic and anaerobic fermentation of *C. angustifolium* leaf extracts, expressed in RE mg/g, *n* = 3, *p* ≤ 0.05.

Sample, Fermentation Time, Drying Temperature, Extraction Solvent	Compound	t_R_, min *	C_RE_, mg/g **
Leaves, 24 h aerobic fermentation, 35 °C, 75% (*v*/*v*) aqueous MeOH	trans-p-coumaric acid	40.74 ± 0.02	18.52 ± 0.05
Leaves, 48 h aerobic fermentation, 35 °C, 75% (*v*/*v*) aqueous MeOH	trans-p-coumaric acid	40.75 ± 0.04	32.38 ± 0.09
Leaves, 72 h aerobic fermentation, 35 °C, 75% (*v*/*v*) aqueous MeOH	trans-p-coumaric acid	40.75 ± 0.02	<0.1 ± 0.01
Leaves, 24 h anaerobic fermentation, 35 °C, 75% (*v*/*v*) aqueous MeOH	trans-p-coumaric acid	40.74 ± 0.02	3.04 ± 0.08
Leaves, 48 h anaerobic fermentation, 35 °C, 75% (*v*/*v*) aqueous MeOH	trans-p-coumaric acid	40.71 ± 0.04	12.07 ± 0.03
Leaves, 72 h anaerobic fermentation, 35 °C, 75% (*v*/*v*) aqueous MeOH	trans-p-coumaric acid	40.78 ± 0.05	<0.1 ± 0.01

* Retention time in min. ** Concentration expressed in Rutin Equivalents (RE) in mg/g.

**Table 9 molecules-29-05530-t009:** Identification and quantification of phenolic compounds in *C. angustifolium* leaf extracts frozen for 3 and 6 months at −18 °C, −80 °C, and −196 °C, *n* = 3, *p* ≤ 0.05.

Sample, Temperature, Storing Time, Extraction Solvent	Compound	t_R_, min *	C_RE_, mg/g **
Leaves, −18 °C, 3 months, 75% (*v*/*v*) aqueous MeOH	Oenothein B	7.14 ± 0.02	41.09 ± 0.11
	Chlorogenic acid	30.52 ± 0.04	35.69 ± 0.09
	Rutin	40.11 ± 0.07	9.78 ± 0.02
Leaves, −80 °C, 3 months, 75% (*v*/*v*) aqueous MeOH	Oenothein B	7.12 ± 0.03	50.11 ± 0.09
	Chlorogenic acid	30.59 ± 0.06	40.41 ± 0.06
	Rutin	40.09 ± 0.09	11.09 ± 0.11
Leaves, −196 °C, 3 months, 75% (*v*/*v*) aqueous MeOH	Oenothein B	7.11 ± 0.04	59.26 ± 0.10
	Chlorogenic acid	30.61 ± 0.03	47.22 ± 0.07
	Rutin	40.12 ± 0.07	14.49 ± 0.05
Leaves, −18 °C, 6 months, 75% (*v*/*v*) aqueous MeOH	Oenothein B	7.12 ± 0.01	34.09 ± 0.12
	Chlorogenic acid	30.55 ± 0.04	30.98 ± 0.11
	Rutin	40.15 ± 0.02	6.99 ± 0.10
Leaves, −80 °C, 6 months, 75% (*v*/*v*) aqueous MeOH	Oenothein B	7.16 ± 0.03	46.08 ± 0.09
	Chlorogenic acid	30.61 ± 0.02	37.49 ± 0.14
	Rutin	40.05 ± 0.01	9.29 ± 0.09
Leaves, −196 °C, 6 months, 75% (*v*/*v*) aqueous MeOH	Oenothein B	7.12 ± 0.04	56.48 ± 0.18
	Chlorogenic acid	30.58 ± 0.08	42.10 ± 0.17
	Rutin	40.10 ± 0.04	13.37 ± 0.16

* Retention time in min. ** Concentration expressed in Rutin Equivalents (RE) in mg/g.

## Data Availability

The primary data summarized in this study are available on request from the corresponding author. The raw data are not publicly available due to low value when not analysed, compared, and summarized.

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
