# Peer review of "Extraction of Bioactive Compounds and Influence of Storage Conditions of Raw Material Chamaenerion angustifolium (L.) Holub Using Different Strategies"

_molecules, 2024, doi:10.3390/molecules29235530_

Round 1

Reviewer 1 Report

Comments and Suggestions for Authors This paper describes the efficiency of various extraction methods and the influence of storage conditions on the quantity of biologically active compounds derived from Chamaenerion angustifolium (L) Holub.   The topic was discussed by other researchers, but in a slightly different context. Therefore, I consider it current and interesting.

In my opinion, the authors presented the methodology in detail and in a comprehensive way. The results are presented in the form of clear tables.

It is my view that the discussion could be more comprehensive, particularly in terms of comparing the results with other relevant works regarding the extraction of Chamaenerion angustifolium (L) Holub under other conditions or using other methods. Thus, the number of references will increase.

I suggest adding a chapter, Concluding Remarks, that summarizes the presented data.

Author Response

Please see the file atta

1st Open Review

  1. I suggest adding a chapter, Concluding Remarks, that summarizes the presented data.

Few studies use different extraction methods, specifically on C. angustifolium. The study aimed to evaluate the best method to extract biologically active compounds using direct extraction with a combination of natural aerobic/anaerobic fermentation. The second part of the study evaluated the influence of storage conditions between 3 different temperatures.

Thank you for your suggestions. The manuscript was supplemented with additional matter, as recommended.

ched 

Reviewer 2 Report

Comments and Suggestions for Authors

Please add the statistical analysis to the results shown in all tables.

Was the plant material botanically identified?

Decrease the similarity percentage of certain sections of the manuscript. For instance, sections 4.4 and 4.5.

In materials and methods add the statistical analysis section.

What does RE mean?

In Table 3, it is mentioned that the fermentation time had an effect on the total flavonoid and phenolic content and radical scavenging activity of the samples. But, as they are displayed, you only show the values between different aqueous concentrations of MetOH.

Author Response

Please see the file attached.

2nd Open Review

  1. Please add the statistical analysis to the results shown in all tables.

All assays were conducted in triplicate (n=3), with results expressed in mean values ± standard deviation. To harmonize results, all values are expressed in rutin equivalents RE mg/g, where calibration graphs were constructed using rutin standard solutions.

The manuscript was supplemented with additional matter, as recommended.

  1. Was the plant material botanically identified?

Yes, the plant was botanically identified using the “Plants of the World Online” database, which searched for “Chamerion angustifolium (L.) Holub” and “Epilobium angustifolium L.”. Based on the stored images, the morphological features of collected plant samples were compared.

  1. Decrease the similarity percentage of certain sections of the manuscript. For instance, sections 4.4 and 4.5.

The similarity percentages in sections 4.4 and 4.5 are decreased. The manuscript was updated accordingly.

  1. In materials and methods add the statistical analysis section.

Section 4.6, “Statistical Analysis”, was added to the manuscript.

  1. What does RE mean?

As described in section 4.4, “Spectrometric Analysis”, all measurements were compared to a standard curve prepared using rutin solution and expressed as rutin equivalents (mean) in mg of rutin standard per 1g of dry raw material: RE mg/g. Additionally, below each table, “** Concentration expressed in rutin equivalents (RE) in mg/g” is added to make it easier for the reader to follow. In first mentioning, Table 1 in the Results section, a description is added “samples expressed in rutin equivalents (RE) per gram of dry material, mg/g”.

  1. In Table 3, it is mentioned that the fermentation time had an effect on the total flavonoid and phenolic content and radical scavenging activity of the samples. But, as they are displayed, you only show the values between different aqueous concentrations of MetOH.

Yes, the natural aerobic fermentation time affected samples by increasing the total content of phenolic compounds, flavonoids and radical scavenging activity. The amount of compounds increased after 24 and 48 hours and then started to decrease after 72 hours of fermentation. In different fermentation periods (24-72h), using the same 0-75% (v./v.)  aqueous MeOH showed that increasing the concentration increases the compound's yield. There were differences in the number of compounds using 35°C and 60°C drying as well. The study aimed to evaluate the best method to achieve maximum efficiency in extracting biologically active compounds, and Table 3's results focus only on the fact that 48h natural aerobic fermentation followed by 75% (v./v.) aqueous MeOH extraction and 35°C drying yields the best efficiency.

Reviewer 3 Report

Comments and Suggestions for Authors

Dear authors,

The manuscript is clearly written and easily understandable.

During reading and analyzing the manuscript I noticed some places in which you should consider corrections.

 My personal suggestions are the following:

- Table 3 - samples leaves, 72h, 60°C are missing in this table. Please check the names of your samples;

- Lines 315-326: should be moved to the Materials and methods section;

- Line 326: correct "Table 7-9" to "Tables 7-9";

- Section 2.4. - Quantitative HPLC-ED analysis: did you analyze the fresh leaves too? As you compared previous results to fresh samples, it would be useful to do in this part too. Why didn't you analyze the blossoms and stems, as you did in previous results? Please give the explanation in your plan of the experiment.

- Line 333: you say that 3 compounds were identified, but, there are 4 compounds given in the Table 7;

- Lines 334-338: you have repeating facts;

- Lines 423-424: which samples do you speak about here?

- Materials and methods: you said that you prepared about 20 ml of methanolic extracts (lines 490-494). Later, you use 100 ml, 80 ml and 77 ml of sample extract solution for determination of phenolics, flavonoids and DPPH. Please correct the methods according to real conditions applied.

 I wish you success!

Author Response

Please see the file attached

3rd Open Review

  1. Table 3 - samples leaves, 72h, 60°C are missing in this table. Please check the names of your samples.

Thank you. That was a typo mistake. Instead of Leaves, Blossoms was typed. The manuscript has been changed according your remark.

  1. Lines 315-326: should be moved to the Materials and methods section.

The manuscript has been changed accordingly. The content was moved to the 4.5 “High-Performance Liquid Chromatography Using Electrochemical Detection” section.

  1. Line 326: correct "Table 7-9" to "Tables 7-9".

The manuscript has been changed accordingly.

  1. Section 2.4. - Quantitative HPLC-ED analysis: did you analyze the fresh leaves too? As you compared previous results to fresh samples, it would be useful to do in this part too. Why didn't you analyze the blossoms and stems, as you did in previous results? Please give the explanation in your plan of the experiment.

Fresh samples were not analyzed with the HPLC-ED system, as the purpose of this analysis was not to compare or identify degradation between fresh and air-dried samples but rather to identify which compounds remain detectable after the drying process. Additionally, blossoms and stems were not included in this study because spectrometric analysis indicated that most biologically active compounds were present in the leaves. Therefore, leaves were chosen as the sole focus for this analysis to ensure a more targeted and relevant investigation of the biologically active compounds.

The explanation of this approach is added to the 4.5 “High-Performance Liquid Chromatography Using Electrochemical Detection” section.

  1. Line 333: you say that 3 compounds were identified, but, there are 4 compounds given in the Table 7.

In Table 7, 4 compounds were identified, but 3 of those were identified, comparing the retention times to external standards. Justification for oenothein B (4th compound) identification is written as follows: “Additionally, previous studies [13] indicated that using reversed-phase HPLC systems with UV detection identified oenothein B being eluted the first from the column in high quantities. The results of this study showed the first compound being eluted from the column in high quantities and that was allocated to oenothein B.”. Additionally, a statement that 4 compounds in total were identified is added to the manuscript.

  1. Lines 334-338: you have repeating facts.

We do not understand the reviewer's response about repeating facts. The information in the reviewer’s lines describes the results of the HPLC-ED analysis. The paragraph was slightly changed to keep the results more informative.

  1. Lines 423-424: which samples do you speak about here?

The samples are as follows: air-dried leave samples, aerobic/anaerobic fermentation leave samples and frozen leave samples. Additional information was added to the manuscript to make it clear.

  1. Materials and methods: you said that you prepared about 20 ml of methanolic extracts (lines 490-494). Later, you use 100 ml, 80 ml and 77 ml of sample extract solution for determination of phenolics, flavonoids and DPPH. Please correct the methods according to real conditions applied.

Thank you for the remark. Yes, 20 ml of methanolic extracts were prepared in the study. The amounts in 4.4 “Spectrometric Analysis” methods were written incorrectly. Some of the method information was deleted to avoid repetition from our previous studies. The manuscript has been updated accordingly, and the correct and real conditions have been applied.

Round 2

Reviewer 2 Report

Comments and Suggestions for Authors

Dear authors,

I suggest you run an ANOVA test and compare either Tuckey's or Dunnet's methods to the results you have. You can't say that your treatments significantly affect your response variables if you do not run a statistical analysis.

Author Response

Dear Reviewer,

Please find attached file with the corrections as suggested.

Sincerely yours, 

Prof. Audrius Maruska
